# Positron emission tomography and magnetic resonance imaging in experimental human malaria to identify organ-specific changes in morphology and glucose metabolism: A prospective cohort study

John Woodford[1,2]*, Ashley Gillman[3], Peter Jenvey[4], Jennie Roberts[4], Stephen Woolley[1,5], Bridget E. Barber[1,6], Melissa Fernandez[1], Stephen Rose[3], Paul Thomas[7], Nicholas M. Anstey[6], James S. McCarthy[1,2]

1 Clinical Tropical Medicine Laboratory, QIMR-Berghofer Medical Research Institute, Brisbane, Australia, 2 University of Queensland, Brisbane, Australia, 3 Commonwealth Scientific and Industrial Research Organisation, Brisbane, Australia, 4 Department of Radiology, Royal Brisbane and Women's Hospital, Brisbane, Australia, 5 Centre for Defence Pathology, Royal Centre for Defence Medicine, Birmingham, United Kingdom, 6 Global and Tropical Health Division, Menzies School of Health Research and Charles Darwin University, Darwin, Australia, 7 Herston Imaging Research Facility, Brisbane, Australia

* john.woodford@uqconnect.edu.au

## Abstract

### Background

*Plasmodium vivax* has been proposed to infect and replicate in the human spleen and bone marrow. Compared to *Plasmodium falciparum*, which is known to undergo microvascular tissue sequestration, little is known about the behavior of *P. vivax* outside of the circulating compartment. This may be due in part to difficulties in studying parasite location and activity in life.

### Methods and findings

To identify organ-specific changes during the early stages of *P. vivax* infection, we performed 18-F fluorodeoxyglucose (FDG) positron emission tomography/magnetic resonance imaging (PET/MRI) at baseline and just prior to onset of clinical illness in *P. vivax* experimentally induced blood-stage malaria (IBSM) and compared findings to *P. falciparum* IBSM. Seven healthy, malaria-naive participants were enrolled from 3 IBSM trials: NCT02867059, ACTRN12616000174482, and ACTRN12619001085167. Imaging took place between 2016 and 2019 at the Herston Imaging Research Facility, Australia. Postinoculation imaging was performed after a median of 9 days in both species ($n = 3$ *P. vivax*; $n = 4$ *P. falciparum*). All participants were aged between 19 and 23 years, and 6/7 were male. Splenic volume (*P. vivax*: +28.8% [confidence interval (CI) +10.3% to +57.3%], *P. falciparum*: +22.9 [CI −15.3% to +61.1%]) and radiotracer uptake (*P. vivax*: +15.5% [CI −0.7% to +31.7%], *P. falciparum*: +5.5% [CI +1.4% to +9.6%]) increased following infection with each species, but more so in *P. vivax* infection (volume: $p = 0.72$, radiotracer uptake: $p = 0.036$). There was no change in

**Data Availability Statement:** Raw imaging files are electronically tagged with personally identifiable information and cannot be shared publicly to protect the confidentiality of participants. All other data are within the manuscript and its Supporting Information files.

**Funding:** This project was supported by a HIRF Seed Funding Grant, Metro North Hospital and Health Service (J.W) and by the Australian National Health and Medical Research Council (J.S.M #1135955, #1037304, #1132975, and N.M.A #1135820, 1098334). The clinical trials contributing participants were funded by the Australian National Health and Medical Research Council (J.S.M #1132975), the Bill and Melinda Gates Foundation (J.S.M OPP1111147) and the Global Health Innovative Technology Fund. The funders had no role in study design, data collection and analysis, decision to publish, or preparation of the manuscript.

**Competing interests:** The authors have declared that no competing interests exist.

**Abbreviations:** CI, confidence interval; FDG, 18-F fluorodeoxyglucose; IBSM, induced blood-stage malaria; iRBC, infected red blood cell; MRI, magnetic resonance imaging; PET, positron emission tomography; qPCR, quantitative polymerase chain reaction; ROI, region of interest; STROBE, Strengthening The Reporting of OBservational Studies in Epidemiology; SUV, standardized uptake value; ULN, upper limit of normal.

FDG uptake in the bone marrow (*P. vivax*: +4.6% [CI −15.9% to +25.0%], *P. falciparum*: +3.2% [CI −3.2% to +9.6%]) or liver (*P. vivax*: +6.2% [CI −8.7% to +21.1%], *P. falciparum*: −1.4% [CI −4.6% to +1.8%]) following infection with either species. In participants with *P. vivax*, hemoglobin, hematocrit, and platelet count decreased from baseline at the time of postinoculation imaging. Decrements in hemoglobin and hematocrit were significantly greater in participants with *P. vivax* infection compared to *P. falciparum*. The main limitations of this study are the small sample size and the inability of this tracer to differentiate between host and parasite metabolic activity.

## Conclusions

PET/MRI indicated greater splenic tropism and metabolic activity in early *P. vivax* infection compared to *P. falciparum*, supporting the hypothesis of splenic accumulation of *P. vivax* very early in infection. The absence of uptake in the bone marrow and liver suggests that, at least in early infection, these tissues do not harbor a large parasite biomass or do not provoke a prominent metabolic response. PET/MRI is a safe and noninvasive method to evaluate infection-associated organ changes in morphology and glucose metabolism.

## Author summary

### Why was this study done?

- In contrast to *Plasmodium falciparum*, blood-stage *Plasmodium vivax* is not traditionally thought to accumulate outside of the circulating blood compartment.

- Emerging data suggest that a hidden compartment of *P. vivax* may exist outside of circulation, in reticulocyte (very young red blood cells)-containing organs.

- The presence of a hidden compartment affects our understanding of the basic biology and pathology of this common parasite and may have implications when developing antimalarial treatments.

- Studying the accumulation of malaria parasites is extremely challenging in vivo and has historically been limited to late stage disease and postmortem studies.

### What did the researchers do and find?

- We performed whole body magnetic resonance imaging (MRI) and positron emission tomography (PET) in individuals undergoing experimental malaria infection to identify changes in organ morphology and glucose metabolism following infection.

- Splenic uptake of radiolabeled glucose increased following *P. vivax* and *P. falciparum* infection and was more pronounced in the *P. vivax* group.

- Glucose uptake in the liver and bone marrow was not increased following infection with either *P. vivax* or *P. falciparum*.

**What do these findings mean?**

- Increased splenic glucose metabolism is present in the early stages of infection with both *P. vivax* and *P. falciparum* and is more pronounced in *P. vivax*, consistent with other emerging evidence of a greater predilection for the spleen in *P. vivax* than *P. falciparum*.

- Functional medical imaging may be a useful tool to study biological processes in experimental malaria infection.

## Introduction

*Plasmodium vivax and Plasmodium falciparum* have key differences in their biology, pathophysiology, and disease manifestations. These are thought to largely arise from differences in microvascular parasite sequestration [1–3] and host red blood cell tropism, with *P. vivax* having a strict tropism for immature reticulocytes [4,5]. Differences in organ-specific parasite tropism have also been proposed [3]. Although *P. vivax* is not thought to undergo significant microvascular tissue sequestration, there is emerging evidence that the spleen may harbor a large hidden reservoir of replicating parasites [6,7]. The evaluation of sequestration and organ-specific parasite tropism has largely relied on biochemical markers, nonhuman primate models, and postmortem studies, which have been critical to progress our understanding of disease pathophysiology but are imperfect tools to study the dynamics of infection in humans.

The tissue distribution of *P. falciparum* on autopsy has been well described. In a seminal publication by Marchiafava and Bignami, the greatest concentration of schizonts was found in the brain, followed by the lungs, spleen, bone marrow, liver, and intestines [1]. Although human autopsy studies of *P. vivax* are comparatively rare, parasite material has been identified in the spleen, liver, and lungs, but not the brain [8]. Studies of *P. vivax* tropism in nonhuman primates have been limited by the need for prior splenectomy [9]. In these studies, few or no parasites were identified in the brain or intestines, whereas the bone marrow and liver sinusoids were major tissue reservoirs for *P. vivax* schizonts and gametocytes [9]. An apparent predilection for the spleen, bone marrow, and liver is consistent with biomarker evidence of a hidden biomass of parasite accumulating in non-endothelial lined spaces [10] and the strict reticulocyte tropism of this species, which may promote concentration in organs in which immature reticulocytes accumulate [4,11]. Nonetheless, this hypothesis is challenging to confirm in life, where these processes are largely inaccessible to evaluation. In this context, magnetic resonance imaging (MRI) and functional imaging using nuclear medicine such as positron emission tomography (PET) may be useful noninvasive methods to study deep tissue processes and organ-specific tropism, particularly in early and non-severe infection.

As access to clinical imaging infrastructure in malaria endemic areas continues to improve, an increasing number of studies have demonstrated the utility of MRI to evaluate the brain in cerebral [12] and uncomplicated falciparum malaria [13,14]. In contrast, PET imaging in malaria is limited to a small number of nonhuman primate studies [15,16] and a single case report of serendipitous imaging in suspected lymphoma [17]. In the nonhuman primate studies, uptake of the glucose biomimetic 18-F fluorodeoxyglucose (FDG) was colocated with sequestered infected red blood cells (iRBCs) as ascertained on subsequent autopsy [16], suggesting that PET imaging using this widely available radiotracer that targets glucose utilization

may be a useful modality to localize organ-specific parasite accumulation and host response in life.

We sought to investigate organ-specific changes in morphology and glucose metabolism in the spleen, vertebral bone marrow, and liver in early *P. vivax* and *P. falciparum* infection using FDG PET/MRI imaging of participants with experimentally induced blood-stage malaria (IBSM). We hypothesized that alterations in glucose metabolism would be present in the regions of interest (ROIs). Secondly, we hypothesized that the patterns of PET/MRI would differ between participants with *P. vivax* and *P. falciparum* infection.

## Methods

This was a prospective, single-center exploratory study performed between 2016 and 2019 at the Herston Imaging Research Facility, Brisbane, Australia. Participants were recruited to the imaging study after enrolling in IBSM studies. In brief, baseline FDG PET/MRI was performed prior to malaria inoculation. Healthy, malaria-naive participants were then inoculated with approximately 564 viable *P. vivax* or approximately 2,800 *P. falciparum* (3D7) infected erythrocytes. Peripheral blood parasitemia was monitored at least daily by quantitative polymerase chain reaction (qPCR) from day 4 after inoculation as described previously [18]. A postinoculation FDG PET/MRI was performed near to the peak of parasitemia, in the 24 hours prior to administration of antimalarial treatment. Standard safety assessments including clinical review, hematology, and biochemistry parameters were performed at specified times during the IBSM studies including baseline, 1 day following postinoculation imaging (at time of treatment), and in convalescence, 1 to 2 weeks after postinoculation imaging [19,20]. A total of 8 participants were planned for enrollment, based on the availability of funding, IBSM study cohorts, and imaging facilities.

### Imaging procedures

Baseline and postinoculation FDG PET/MRI were performed on all participants using the 3 Tesla Magnetom Biograph mMR (Siemens, Erlangen, Germany). All participants completed an institutional MRI safety checklist prior to imaging. The female participants underwent qualitative human chorionic gonadotropin testing prior to imaging to exclude pregnancy. Participants were required have a normal fasting glucose at enrolment and to abstain from strenuous exercise and follow a low-carbohydrate diet in the 24 hours prior to imaging with a 6-hour fasting period immediately prior to help standardize FDG uptake [21,22]. Diet and activity information sheets were provided to assist participant adherence. Due to the different growth rate of parasites in IBSM, the interval between baseline and postinoculation imaging differed in the *P. vivax* and *P. falciparum* studies.

Nuclear medicine and radiology specialists reporting all imaging were not aware of the inoculum species. Mean standardized uptake values (SUVs) were estimated for the prespecified ROIs including the spleen, vertebral bone marrow, and liver. SUV is the most commonly used measurement of radiotracer activity and represents the ratio of tissue radioactivity related to FDG at a given time and the injected dose of radioactivity per kilogram of participant weight. Organ volumes were calculated for the spleen and liver. As an exploratory analysis, the quantitative rate of irreversible radiotracer uptake was modeled for the ROIs. Detailed methods and imaging protocols are outlined in the S1 Text.

### Statistical analysis

All imaging results are presented descriptively for the *P. vivax* and *P. falciparum* groups. Where relevant, results for each inoculum group are presented as mean (95% confidence

interval (CI)) percentage change from baseline for quantitative imaging metrics. A >10% percentage difference in SUV from baseline to postinoculation imaging was considered meaningful based on intraindividual, inter-scan variability reported in stringent FDG PET imaging trials in malignancy and non-malignancy settings [23–25]. Parasitemia at the time of postinoculation imaging was estimated by interpolating the linear rate of change between the nearest log-transformed study parasitemia measurements.

Statistical analysis was performed using Prism 8.4.2 (GraphPad, San Diego, United States of America). In an exploratory analysis, baseline and postinoculation imaging metrics and selected hematology and biochemistry parameters were compared using paired, 2-tailed *t* tests. Differences between the mean percentage change from baseline for quantitative imaging metrics for *P. vivax* and *P. falciparum* groups were compared using unpaired *t* tests. Differences between selected hematology and biochemistry parameters and parasitemia measurements for *P. vivax* and *P. falciparum* groups were compared using unpaired *t* tests of Mann–Whitney U tests. A *p*-value <0.05 was considered significant. No adjustments have been made for multiple comparisons.

### Study approval

Participants were recruited from 3 IBSM studies: 2 published (*P. falciparum*: NCT02867059 [26] and *P. vivax*: ACTRN12616000174482 [27]) and 1 unpublished (*P. falciparum*: ACTRN12619001085167). These studies were approved by the QIMR-Berghofer Medical Research Institute Ethics Committee. The exploratory imaging study was approved by the QIMR-Berghofer Medical Research Institute and Queensland Health Metro North Ethics committees. The exploratory imaging study was prospectively linked to 2 of the IBSM studies (ACTRN12616001458426 linked to IBSM study NCT02867059 and ACTRN12616001238460 linked to IBSM study ACTRN12616000174482 (see S1 Protocol)). The third IBSM study incorporated the exploratory imaging study into the main protocol (ACTRN12619001085167). All participants provided written informed consent for the IBSM studies and the exploratory imaging study. This study has been reported as per the Strengthening The Reporting of OBservational Studies in Epidemiology (STROBE) guidelines (S1 Checklist).

## Results

A total of 8 healthy, malaria-naive participants were recruited. All participants underwent baseline imaging within 7 days before inoculation. One participant was enrolled and underwent baseline imaging, but was excluded from the IBSM study before inoculation due to incidental hypokalemia on baseline biochemistry. This participant did not undergo postinoculation imaging and was excluded from the analysis. Seven participants underwent postinoculation imaging. The study population is described in Table 1. Included participants had a median age of 20 years (range 19 to 23), and 6/7 were male. Three were inoculated with *P. vivax* and 4 with *P. falciparum*. All participants developed a measurable parasitemia (Fig 1) and reported symptoms of early malaria infection prior to treatment. Parasitemia at postinoculation imaging was at least as high in the *P. falciparum* group (median 22,326 [range 998 to 99,134 parasites/mL]) as the *P. vivax* group (median 6,042 [range 5,818 to 29,097 parasites/mL, Mann–Whitney U test *p* > 0.99]). Although the day of treatment varied between studies, both groups underwent imaging a median 9 days following inoculation. Imaging results for the ROIs including organ volumes and SUVs are presented in Fig 2 and S1 Table. Exploratory calculation of the quantitative rate of radiotracer uptake for the ROIs is presented in the S1 Table and S2 Text.

**Table 1. Study population information.**

| Participant | Age (years) | Sex | BMI | Trial | Challenge agent | Dose (viable p/mL) | Day of postinoculation imaging | Parasitemia at imaging (p/mL) |
|---|---|---|---|---|---|---|---|---|
| 1 | 20 | F | 29.5 | Collins et al. (2020) [27] | Pv | 564 | 9 | 29,097 |
| 2 | 23 | M | 22.9 | Collins et al. (2020) [27] | Pv | 564 | 9 | 5,818 |
| 3 | 20 | M | 22.0 | Collins et al. (2020) [27] | Pv | 564 | 9 | 6,042 |
| 4 | 19 | M | 28.5 | Gaur et al. (2020) [26] | Pf | 2,800 | 7 | 1,427 |
| 5 | 22 | M | 22.0 | Gaur et al. (2020) [26] | Pf | 2,800 | 7 | 998 |
| 6 | 19 | M | 19.7 | ACTRN12619001085167 | Pf | 2,800 | 11 | 43,224 |
| 7 | 19 | M | 24.2 | ACTRN12619001085167 | Pf | 2,800 | 10 | 99,134 |

BMI, body mass index; Pf, *P. falciparum*; Pv, *P. vivax*.

Parasitemia at imaging: estimated by interpolating the linear rate of change between the closest log-transformed parasitemia measurements.

## Splenic imaging

In the *P. vivax* group, splenic SUVs increased +15.5% from baseline (CI −0.7% to +31.7%). SUVs on postinoculation imaging were significantly higher on postinoculation imaging compared to baseline imaging (paired *t* test *p* = 0.044). This was visible on qualitative review in some participants (Fig 3). Splenic volume increased a mean +28.8% from baseline imaging but did not reach statistical significance (CI +0.3% to +57.3%, paired *t* test *p* = 0.053).

In the *P. falciparum* group, splenic SUVs increased +5.5% from baseline (CI +1.4% to +9.6%, paired *t* test *p* = 0.019), less than the prespecified 10% threshold. Splenic volume increased +22.9% from baseline but did not reach statistical significance (CI −15.3% to +61.1%, paired *t* test *p* = 0.20.) Participant 4 demonstrated a slight reduction in splenic volume, contrary to all other participants.

Compared to the *P. falciparum* group, the *P. vivax* group demonstrated a greater increase in splenic SUV from baseline to postinoculation imaging (+15.5% [CI −0.7% to +31.7%] versus +5.5% [CI +1.4% to +9.6%], unpaired *t* test *p* = 0.036). The change in splenic volume was larger in the *P. vivax* group compared to the *P. falciparum* group but did not reach statistical significance (+28.8% [CI +0.3% to +57.3%] versus +22.9% [CI −15.3% to +61.1%], unpaired *t* test *p* = 0.72).

## Vertebral bone marrow imaging

Vertebral bone marrow SUVs were stable on postinoculation imaging in both the *P. vivax* group (+4.6% [CI −15.9% to +25.0%], paired *t* test *p* = 0.44) and the *P. falciparum* group

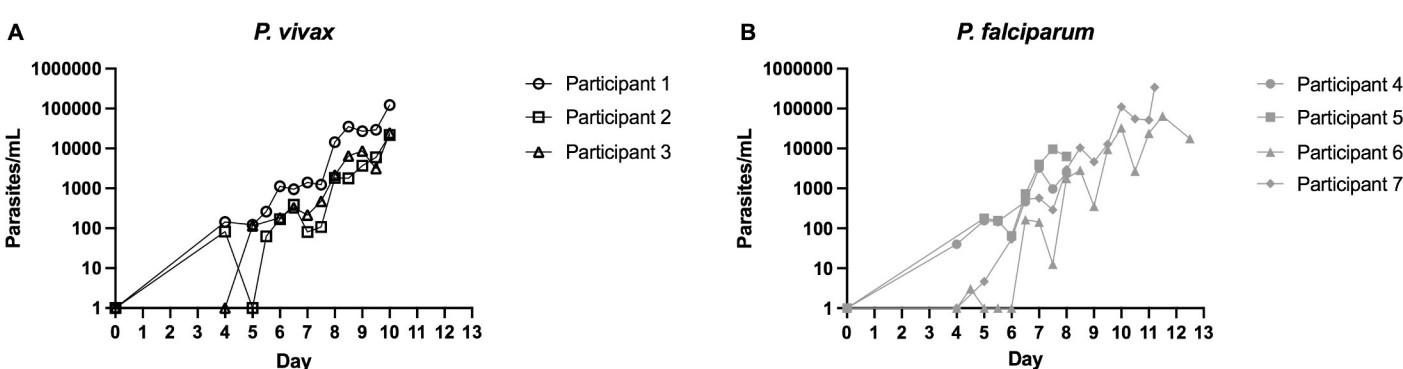

**Fig 1.** Parasitemia curves for participants inoculated with **(A)** *P. vivax* and **(B)** *P. falciparum*.

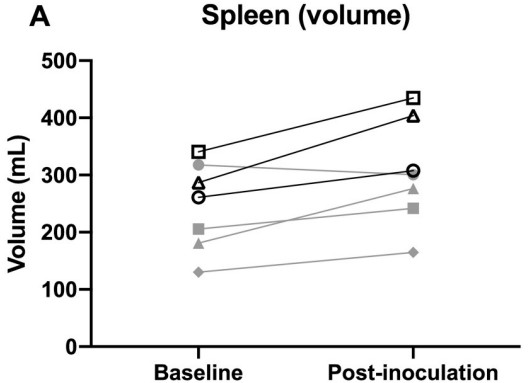

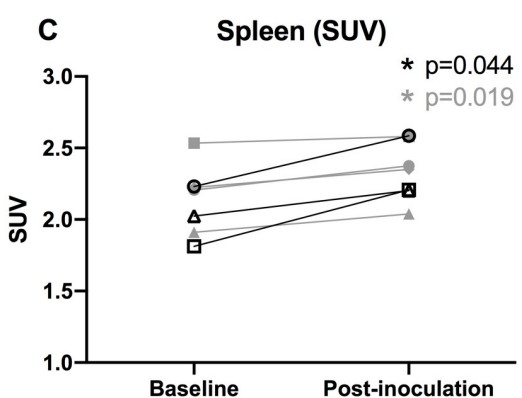

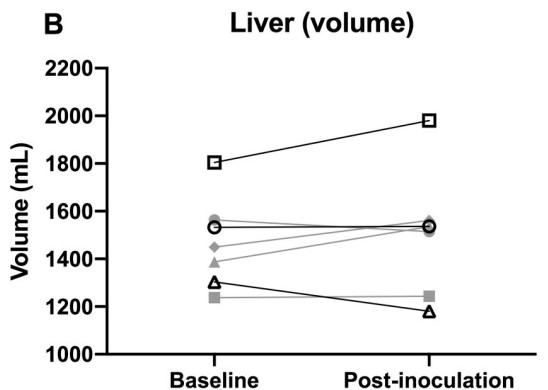

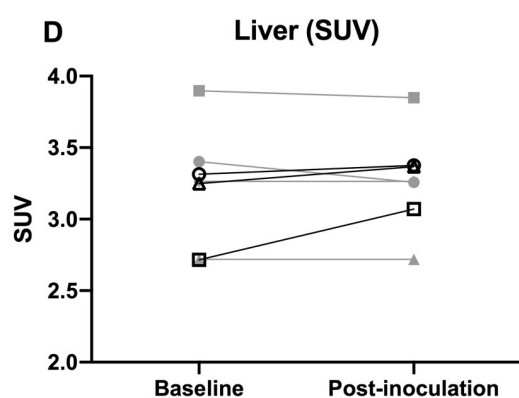

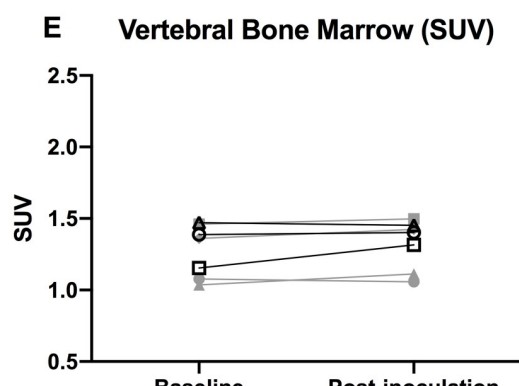

**Fig 2. Imaging indices at baseline and postinoculation. (A)** Spleen (volume), **(B)** Liver (volume), **(C)** Spleen (SUV), **(D)** Liver (SUV), and **(E)** Vertebral Bone Marrow (SUV). *P. vivax* group represented by black lines with unfilled markers. *P. falciparum* group

represented by gray lines with filled markers. Asterisk represents $p < 0.05$ in paired $t$ test comparison with baseline measurements (black for *P. vivax*, gray for *P. falciparum*). SUV, standardized uptake value.

(+3.2% [CI −3.2% to +9.6%], paired $t$ test $p = 0.17$). Participant 2 (*P. vivax* group) was the only individual demonstrating a change in SUV above the prespecified 10% threshold. There was no significant difference in change in percentage change in SUV between the groups (unpaired $t$ test $p = 0.77$).

## Hepatic imaging

In both the *P. vivax* and *P. falciparum* groups, hepatic volume and SUVs remained stable on postinoculation imaging (*P. vivax*: hepatic volume: +0.2% [CI −23.6% to +24.0%], paired $t$ test $p = 0.84$; SUV: +6.2% [CI −8.7% to +21.1%], paired $t$ test $p = 0.18$; *P. falciparum*: hepatic volume: +3.9% [CI −6.2% to 14.0%], paired $t$ test $p = 0.32$; SUV: −1.4% [CI −4.6% to +1.8%], paired $t$ test $p = 0.25$). Participant 2 (*P. vivax* group) was the only individual demonstrating a change in SUV above the prespecified 10% threshold.

There was no significant difference in hepatic volume change or SUV change between the groups (percentage change in volume unpaired $t$ test $p = 0.56$ and SUV unpaired $t$ test $p = 0.06$).

## Safety, clinical review, and laboratory parameters

No imaging-related adverse events or incidental radiological findings requiring clinical follow-up were recorded. All participants experienced symptoms of early malaria infection. Non-tender palpable splenomegaly was reported in Participant 3 following *P. vivax* inoculation on routine examination performed on the day of postinoculation imaging. This was present on inspiration, extended 1 cm below the costal margin, and resolved on follow-up examination 3 days following postinoculation imaging. Clinical hepatomegaly or splenomegaly was not reported for the remaining participants in either group.

Routine hematology (hemoglobin, hematocrit, and platelet count) and biochemistry (total bilirubin and alanine transaminase) measurements were collected at baseline, postinoculation (at time of treatment), and in convalescence for all participants. The majority of hematology and biochemistry measurements were within the laboratory reference range (S1 Fig, S2 Table).

## Baseline  Post-inoculation

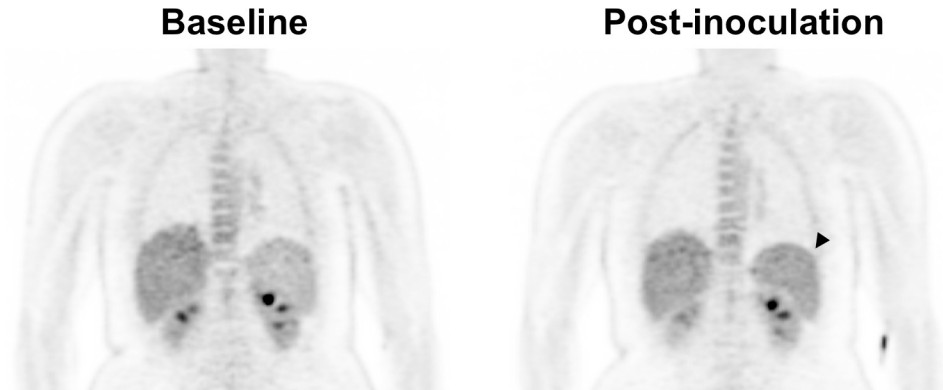

**Fig 3. Baseline and postinoculation PET imaging in Participant 1 (*P. vivax* group).** Increased radiotracer uptake is evident in the spleen on postinoculation imaging (arrowhead). PET, positron emission tomography.

Changes in laboratory parameters in our study population were similar to those observed in the overall contributing IBSM study populations [26,27].

In the *P. vivax* group, hemoglobin and hematocrit decreased significantly between baseline and postinoculation (mean 148 g/L [CI 103 to 193] versus 137 g/L [CI 94 to 180], paired *t* test *p* = 0.045 and 0.43 L/L [CI 0.33 to 0.53] versus 0.39 L/L [CI 0.28 to 0.50], paired *t* test *p* = 0.008, respectively) and further at convalescence (148 g/L [CI 103 to 193] versus 133 g/L [CI 85 to 182], paired *t* test *p* = 0.004 and 0.43 L/L [CI 0.33 to 0.53] versus 0.38 L/L [CI 0.25 to 0.51], paired *t* test *p* = 0.020). Platelets decreased at postinoculation (206 × 10$^9$/L [CI 172 to 241] versus 134 × 10$^9$/L [CI 109 to 159], paired *t* test *p* = 0.034) and increased slightly compared to baseline at convalescence, but did not reach statistical significance (206 × 10$^9$/L [CI 172 to 241] versus 218 × 10$^9$/L [range 211 to 226], paired *t* test *p* = 0.28). There were no statistically significant changes in biochemistry measurements between baseline and postinoculation or between baseline and convalescence (S2 Table), although Participant 1 experienced a clinically relevant elevation in alanine transaminase in convalescence (Day 17, 86 IU/L, 2.2 ×upper limit of normal (ULN)). In the *P. falciparum* group, there were no statistically significant changes in hematology or biochemistry measurements between baseline and postinoculation or between baseline and convalescence (S2 Table). Participants 5, 6, and 7 each experienced elevated alanine transaminase measurements in convalescence (85 IU/L, 2.1 ×ULN; 73 IU/L, 1.8 ×ULN; 128 IU/L, 3.2 ×ULN, respectively).

Compared to the *P. falciparum* group, the hemoglobin decrease was greater from baseline to postinoculation in the *P. vivax* group (mean −7.2% [CI −13.4% to −1.1%] versus +1.2% [CI −1.3% to +3.8%], unpaired *t* test *p* = 0.003). This was similar for hematocrit (−8.7% [CI −13.5% to −3.9%] versus −2.3% [CI −6.4% to +1.9%], unpaired *t* test *p* = 0.017). Conversely, decrease in platelet count was greater from baseline to convalescence in the *P. falciparum* group compared to the *P. vivax* group (−33.5% [CI −69.5% to +2.5%] versus +5.9% [CI −10.7% to +22.6%], unpaired *t* test *p* = 0.035). There were no other differences between the groups in any other changes in hematology or biochemistry measurements.

## Discussion

We demonstrate that medical imaging using PET and MRI are technically feasible and safe in experimental human *P. vivax* and *P. falciparum* infection and can define organ-specific physiological changes very early in disease. The increase in splenic volume and utilization of glucose on postinoculation imaging was most pronounced in the *P. vivax* group. Although the sample size is small, in the exploratory statistical comparisons, increase in splenic SUV was significantly greater in participants with *P. vivax* compared to those with *P. falciparum*. These findings suggest splenic tropism with both species early in the course of disease, but that it is significantly greater for *P. vivax* compared to *P. falciparum*.

Changes in splenic imaging metrics may be associated with either increased parasite or host activity, or a combination of both. This is consistent with nonhuman primate *Plasmodium coatneyi* infection studies, where increased splenic FDG uptake associated with accumulation of iRBCs in the red pulp and lymphoid hyperplasia have been reported [16]. In vitro studies have shown that iRBCs consume up to 100 times the amount of glucose than that consumed by uninfected erythrocytes [28] and that trophozoite uptake is up to 6-fold greater than that of ring stages [29]. While no *Plasmodium* studies have been performed to identify if this in vitro glucose consumption is sufficient to be detected by FDG PET, the principle of clinically detectable tracer uptake by pathogens has been demonstrated in bacteria [30]. Taken together, these findings may suggest that splenic FDG signal reflects iRBC accumulation within the spleen.

The spleen has been proposed as a site for accumulation of *P. vivax* infection [3,6,10,31]. The extent to which this occurs in early infection and in clinical disease is not clear. Indirect evidence points to accumulation of non-phagocytosed *P. vivax* iRBCs in the spleen in acute malaria [4,10], with supporting direct evidence from case reports [6,32]. Furthermore, early splenic rupture may occur more frequently in *P. vivax* compared to *P. falciparum* infection [33], which may suggest a greater tropism of this species for the spleen. In an ex vivo perfused spleen model, a subpopulation of *P. falciparum* ring stage parasites are retained in the red pulp of the spleen [34], hypothesized as mechanism to reduce acute host morbidity and prolong the duration of transmissible infection [35]. In a nonhuman primate *P. vivax* model, spleen-dependent expression of molecules mediating adhesion to human splenic fibroblasts has been described [36], suggesting the presence of a parasite phenotype that may favor splenic accumulation. Finally, very recent data suggest that immature reticulocytes accumulate in the human spleen providing a favorable niche for *P. vivax* accumulation [11]. We now show evidence supporting splenic tropism and splenic accumulation of *P. vivax* very early in infection.

Both *P. falciparum* [1,37] and *P. vivax* [9,38] have also been shown to infect the bone marrow. The lack of signal change in the vertebral bone marrow suggest that this tissue may have a lower parasite biomass in early infection, or alternatively, that parasites located in the bone marrow do not provoke as prominent host metabolic response. This is in keeping with a recent study suggesting that in acute vivax malaria, the parasite concentration in the bone marrow is, on average, no higher than that circulating in peripheral blood [38].

Similarly, while *P. falciparum* [1] and *P. vivax* [8] have been found in liver at autopsy, the absence of volume or tracer signal change in our study suggests that this may not be a site of accumulation early in infection. In our cohort, participants from both the *P. vivax* and *P. falciparum* groups experienced elevated transaminases in convalescence. Elevated transaminases following treatment have been previously described in experimental *P. vivax* [39], *P. falciparum* [40] infections, and in naturally acquired infection in non- or pauci-immune patients [39,41] and are unlikely to be related to imaging, but rather may be related to inflammation and hemolysis [39].

The *P. vivax* group experienced a greater decrease in hematology indices from baseline to postinoculation compared to the *P. falciparum* group. The greater removal of RBCs from circulation previously described in *P. vivax* infection compared to *P. falciparum* infection is thought to occur in the spleen [42], which may also contribute to the imaging findings. Reduction in platelets in experimental malaria has been previously described and may relate to splenic pooling or more generalized platelet removal from circulation due to endothelial activation or parasite killing [43]. We and others have previously described the association between endothelial activation and thrombocytopenia in experimental malaria [44,45], and labeled, auto-transfused platelets have been observed to undergo disseminated extra-splenic consumption in uncomplicated *P. falciparum* infection [46].

There were several limitations to this study. The population of the study was small, and statistical comparisons of baseline imaging to postinoculation imaging are necessarily of an exploratory nature, as there is limited power to discriminate between groups. Secondly, while FDG is a readily available radiotracer, it is relatively nonspecific and is unable to differentiate between host and parasite activity. Advances in radiochemistry may permit the development of a radiotracer with sufficient specificity to selectively image iRBCs. Finally, we were unable to control for other variables that may have influenced the differences observed between the groups such as parasitemia at the time of imaging.

To date, the ability to localize the pathology of malaria in life has been limited, and direct evaluation of sequestration and organ-specific parasite tropism has relied upon animal models and postmortem studies. Using the IBSM model, we have demonstrated that the medical

imaging techniques MRI and PET may be used to study human *P. vivax* and *P. falciparum* infection in life and observe early infection in a way not previously possible. The presence of alterations in splenic glucose uptake and volume soon after blood-stage infection compared to baseline suggests that these imaging techniques may be useful to localize disease. The differences observed between species consolidates the concept of species-specific tissue tropism, where *P. vivax* has a greater predilection for the spleen. Our findings suggest that the hypothesized endosplenic life cycle of *P. vivax* may be established soon after blood-stage infection.

## Supporting information

**S1 Checklist. STROBE Checklist.**
(PDF)

**S1 Text. Methods.**
(PDF)

**S2 Text. Results.**
(PDF)

**S1 Table. Abdominal quantitative imaging metrics.**
(PDF)

**S2 Table. Hematology and biochemistry parameters.**
(PDF)

**S1 Fig. Hematology and biochemistry parameters at baseline, postinoculation, and convalescence. (A)** Hemoglobin (g/L), **(B)** Hematocrit (L/L), **(C)** Platelets (10^9/L), **(D)** Total Bilirubin (mmol/L), and **(E)** ALT (IU/L).
(PDF)

**S1 Protocol. Study protocol.**
(PDF)

## Acknowledgments

We would like to thank Louise Campbell and Peta Gray from HIRF for their help in coordinating the imaging of participants and Dr. Manoj Bhatt at the Department of Nuclear Medicine, RBWH for serving as the imaging medical monitor; Marita Prior at the Department of Radiology, RBWH for her help in coordinating the reporting of imaging; study staff at Q-Pharm Pty Ltd for their help in coordinating the IBSM participant exploratory study involvement; Medicines for Malaria Venture for encouraging investigator-initiated studies; and all participants for volunteering their time.

## Author Contributions

**Conceptualization:** John Woodford, Nicholas M. Anstey, James S. McCarthy.

**Data curation:** John Woodford, Ashley Gillman.

**Formal analysis:** John Woodford, Ashley Gillman, Peter Jenvey, Jennie Roberts.

**Funding acquisition:** John Woodford, James S. McCarthy.

**Investigation:** John Woodford, Stephen Woolley.

**Methodology:** John Woodford, Ashley Gillman, Peter Jenvey.

Project administration: John Woodford, Stephen Woolley, Bridget E. Barber, Melissa Fernandez.

Resources: Melissa Fernandez.

Software: Ashley Gillman.

Supervision: Bridget E. Barber, Stephen Rose, Paul Thomas, Nicholas M. Anstey, James S. McCarthy.

Writing – original draft: John Woodford.

Writing – review & editing: John Woodford, Ashley Gillman, Stephen Woolley, Bridget E. Barber, Melissa Fernandez, Stephen Rose, Paul Thomas, Nicholas M. Anstey, James S. McCarthy.

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
