## [Editor Report · Decision Letter 0]

25 Sep 2020

Dear Dr Woodford, 

Thank you for submitting your manuscript entitled "Positron emission tomography and magnetic resonance imaging in experimental human malaria: greater early splenic tropism in Plasmodium vivax than Plasmodium falciparum infection" for consideration by PLOS Medicine.

Your manuscript has now been evaluated by the PLOS Medicine editorial staff as well as by an academic editor with relevant expertise and I am writing to let you know that we would like to send your submission out for external peer review.

Kind regards,

Artur Arikainen,

Associate Editor

PLOS Medicine

---

## [Decision Letter · Decision Letter 1]

23 Oct 2020

Dear Dr. Woodford,

Thank you very much for submitting your manuscript "Positron emission tomography and magnetic resonance imaging in experimental human malaria: greater early splenic tropism in Plasmodium vivax than Plasmodium falciparum infection" (PMEDICINE-D-20-04576R1) for consideration at PLOS Medicine. 

[LINK]

In light of these reviews, I am afraid that we will not be able to accept the manuscript for publication in the journal in its current form, but we would like to consider a revised version that addresses the reviewers' and editors' comments. Obviously we cannot make any decision about publication until we have seen the revised manuscript and your response, and we plan to seek re-review by one or more of the reviewers. 

We expect to receive your revised manuscript by Nov 13 2020 11:59PM. Please email us (plosmedicine@plos.org) if you have any questions or concerns.

We look forward to receiving your revised manuscript. 

Sincerely,

Artur Arikainen, 

Associate Editor 

PLOS Medicine

plosmedicine.org

1. Please address the reviewers’ comments below.

2. Please revise your title according to PLOS Medicine's style. Your title must be nondeclarative and not a question. It should begin with main concept if possible. "Effect of" should be used only if causality can be inferred, i.e., for an RCT. Please place the study design ("A randomized controlled trial," "A retrospective study," "A modelling study," etc.) in the subtitle (ie, after a colon).

3. Abstract:

a. Please include summary participant demographics (age, sex, etc.).

b. Please include study dates and setting.

c. Please quantify all results with p values and 95% CIs.

d. In the last sentence of the Abstract Methods and Findings section, please describe the main limitation(s) of the study's methodology.

e. Line 54: Please begin with “In this study, we observed that…”

f. Please include trial registration numbers for the included studies.

5. Methods: Please also include trial registration numbers for the two published studies.

6. Line 215: Please avoid “approaching statistical significance” – instead state “was not statistically significant”.

7. Please include as Supporting Information the full protocol for this specific analysis/sub-study, as approved by the ethics committee prior to the start of the study.

8. Please ensure that the study is reported according to, for example, the STROBE or TREND guidelines (which can be found here: http://www.equator-network.org/), and include the completed checklist as Supporting Information. When completing the checklist, please use section and paragraph numbers, rather than page numbers. Please add the following statement, or similar, to the Methods: "This study is reported as per the ___ guideline (S1 Checklist).

---

Comments from the reviewers:

Reviewer #1: The authors report a very interesting and singular report using PET imaging to generate evidence of a relatively high degree of tropism for splenic tissues by P. vivax compared to P. falciparum. 

Some concerns:

1) The authors seem to presume readers understand PET imaging technology. The introduction should make clear precisely what is being detected, i.e., increased glucose consumption per se rather than any parasite-specific signal. The greatest uncertainty appears barely acknowledged -- that PET signal from the spleen may not be parasite biomass, but simply downstream physiological responses to the infection unrelated to biomass in a quantitative sense.

2) The authors cite the specific findings of Barber et al. 2015 (ref.7) that suggested a dominant biomass seated in extravascular spaces, and then express of those, "Nonetheless, this hypothesis is challenging to confirm...". With specific regard to that hypothesis, were Barber et al. the first to propose it? Those authors seem to be point to the CMR review by Baird in 2013 as its origin, as do others. 

3) There appears to have been a merging error as the legend to Figure 3 does not appear in the PDF. The difference between baseline and post-inoculation appears wholly unremarkable to this reviewer's untrained eye. 

4) Reference 3 seems quite important here but is not yet available and, so far as one may now suppose, may not be available in the near future. Are the authors supposing acceptance and full citation by the time this paper gets to press? If so, the details of that investigation should appear in this manuscript. It is not reasonable to cite it as though available, but to withhold the essential findings. 

5) The most compelling evidence of extavascular tropism for P. vivax is not described or cited by the authors, appearing to be a significant omission. Specifically, the multiple papers from the laboratories of Bruce Russell, Benoit Malleret, Laurent Renia, and Wei Hong Tham, especially their Science paper of 2018. If P. vivax invasion occurs only in the youngest reticulocytes (and erythroblasts) expressing CD71, an infection of extravascular hemopoietic tissues (like those in the marrow and spleen) may logically be where the infectious biomass is seated. The exclusion of direct molecular evidence in the context of the original hyposthesis of extravascular tropism is unfortunate. 

Reviewer #2: A hot topic in contemporary malaria research is whether there are hidden life cycles of malaria parasites in certain organs. Discussed sites, where hidden life cycles might occur, include the bone marrow and the spleen. However, conclusive experimental data in support of hidden life cycles are scarce. Woodford et al. have now used an elegant in vivo, whole body imaging techniques to detect organs in which the glucose biomimetic [18F] fluorodeoxyglucose accumulates during the course of experimentally induced P. falciparum and P. vivax blood stage infections in human volunteers. The data presented suggest a strong splenic tropism for P. vivax and, to a lesser extent, for P. falciparum, but no accumulation in the bone marrow or the liver. These findings are interpreted in terms of a putative P. vivax replicative cycle in the spleen.

The selling point of the manuscript is the combination of the imaging technique with the experimentally induced blood stage infection in human volunteers. The observations made are potentially of interest, albeit they do not exclude the possibility of hidden life cycles occurring in organs other than the spleen at later time points in the infection, respectively at higher parasitemia.

Queries:

1. This reviewer assumes that the standard uptake value refers to the uptake of [18F] fluorodeoxyglucose. This should be better explained in the manuscript. Can the SUV be converted into absolute molecule numbers?

2. The different host cell tropism of P. falciparum (mature erythrocytes) and P. vivax (reticulocytes) should be mentioned.

3. The number of volunteers is quite small, raising concerns regarding the statistical significance of the findings.

4. Is the enhanced splenic activity due to replicating parasites or is it a reflection of increased immunological activity, e.g., by macrophages present in the splenic cord, which remove lysed infected reticulocytes and/or pitted parasites following their destruction at the splenic slits?

5. The text could be substantially shortened and the number of display items could be reduced to two. 

Reviewer #3: See attachment

Michael Dewey

Reviewer #4: General comments:

This is an interesting pilot study demonstrating the value of performing imaging in controlled human malaria infection studies. Despite the fact that the number of volunteers in the study is extremely small, the pioneering nature of the research should be applauded. 

However, the main conclusion of the manuscript "greater early splenic tropism in Plasmodium vivax than Plasmodium falciparum infection" seems to be driven primarily by a significant different change in SUV between Pf and Pv inoculated volunteers. Here, there seems to be an error in the analysis. Line 216 states that splenic SUV for the Pv group increases a median of 15.9% (range 8.8-21.8%) whereas line 227 states that splenic SUV in the Pv group increased a median of 27.6% (range 18.0-40.8%). Given this discrepancy, it seems that, comparing with data from table S1, the data from the splenic volume change in the Pv group (median 27.6%) have been accidentally swapped with the data from splenic SUV change (median 21.8%). This should be corrected. Preferably, the statistical tests used for these data should be listed in the results section.

Looking at the graphs presented in figure 2, it seems to me that the difference between splenic SUV change in Pf and Pv inoculated volunteers is mainly driven by one volunteer in the Pf group who had a high baseline SUV and thus did not further increase. Given the high baseline SUV, this volunteer might significantly bias data.

Furthermore, I feel that the clinical review and laboratory parameters could have included more data than just the subset of volunteers for which scans were available. It does not make sense to exclusively present clinical and laboratory data from only the 7 individuals in this paper only, whilst much more data is available from the referenced studies. I would recommend expanding this section to include more data from all participants in the studies.

Lastly, a minor comment regarding the liver enzyme elevations which were detected. Given the timelapse between imaging and enzyme abnormalities (1-2 weeks), it does not seem justified to draw any conclusions on the etiology based on imaging and enzymes correlations.

[LINK]

---

## [Decision Letter · Decision Letter 2]

29 Jan 2021

Dear Dr. Woodford,

Thank you very much for re-submitting your manuscript "Positron emission tomography and magnetic resonance imaging in experimental human malaria: a prospective cohort study to identify organ-specific changes in morphology and glucose metabolism" (PMEDICINE-D-20-04576R2) for review by PLOS Medicine.

I have discussed the paper with my colleagues and the academic editor and it was also seen again by three reviewers. I am pleased to say that provided the remaining editorial and production issues are dealt with we are planning to accept the paper for publication in the journal.

[LINK]

We look forward to receiving the revised manuscript by Feb 05 2021 11:59PM.   

Sincerely,

Artur Arikainen, 

Associate Editor 

PLOS Medicine

plosmedicine.org

Requests from Editors:

1. Title: Please update to: “Positron emission tomography and magnetic resonance imaging in experimental human malaria to identify organ-specific changes in morphology and glucose metabolism: A cohort study”

2. Data Availability Statement: On the submission form you answer “No - some restrictions will apply” followed by “All relevant data are within the manuscript and its Supporting Information files.” Please clarify whether or not there is any restriction on data sharing, and update the statement to be consistent.

3. Please include line numbers in your margin throughout.

4. Abstract: Please quantify the results presented (including p values and 95% CIs): “There was no change in FDG uptake in the bone marrow or liver following infection with either species. In participants with P. vivax, hemoglobin, hematocrit, and platelet count decreased from baseline at the time of post-inoculation imaging. Decrements in hemoglobin and hematocrit were significantly greater in participants with P. vivax infection compared to P. falciparum.”

5. Author Summary: Please clarify “sequester”, “extravascular”, “reticulocyte”, “pharmacokinetic”, and “tropism” for a lay reader, or replace with simpler terms.

6. Please remove spaces from within citations, eg: “…for immature reticulocytes [4,5].”

7. Table 1: The terms gender and sex are not interchangeable (as discussed in http://www.who.int/gender/whatisgender/en/ ); please use the appropriate term.

8. References:

a. 5: Remove “*… *”, and remove “(New York, NY)”

b. 7 and 39: Papers cannot be listed in the reference list until they have been accepted for publication or are otherwise publicly accessible (for example, in a preprint archive). The information may be cited in the text as a personal communication with the author if the author provides written permission to be named. Alternatively, please provide a different appropriate reference. 

c. Please double check journal short names for accuracy/consistency.

9. When completing the STROBE checklist, please use section and paragraph numbers, rather than page numbers.

---

Comments from Reviewers:

Reviewer #1: The authors have done a thorough job of revision, addressing any concerns this reviewer had. This is a very interesting study of merit. 

Reviewer #2: The authors have substantially improved the manuscript. All my comments have been addressed. 

Reviewer #3: The authors have addressed all my points.

Michael Dewey

[LINK]

---

## [Editor Report · Decision Letter 3]

10 Feb 2021

Dear Dr. Woodford,

Thank you very much for re-submitting your manuscript "Positron emission tomography and magnetic resonance imaging in experimental human malaria to identify organ-specific changes in morphology and glucose metabolism: a prospective cohort study" (PMEDICINE-D-20-04576R3) for review by PLOS Medicine.

I have discussed the paper with my colleagues and the academic editor and it was also seen again by xxx reviewers. I am pleased to say that provided the remaining editorial and production issues are dealt with we are planning to accept the paper for publication in the journal.

[LINK]

We look forward to receiving the revised manuscript by Feb 17 2021 11:59PM.   

Sincerely,

Beryne Odeny

Associate Editor 

PLOS Medicine

plosmedicine.org

Requests from Editors:

-Reference 11 listed as under review, papers cannot be listed in the reference list until they have been accepted for publication or are otherwise publicly accessible (for example, in a preprint archive). The information may be cited in the text as a personal communication with the author if the author provides written permission to be named. Alternatively, please provide a different appropriate reference. If you opt to text as a personal communication, please provide the name of the individual, the affiliation, and date of communication. The individual named must provide written permission to be named.

-Please include legends in the Supporting Information files (Tables/Figure)

-In the main text, we cannot locate where the quantitative/statistical results have been reported for the blood/biochemistry results (seems to be reported in text for Pv group but not Pf). 

-The statistical reporting in Figure S1 is not clear. Please add the results for the comparisons to the figure or place the results in a table if they are not in the text as they need to be reported.

Comments from Reviewers:

[LINK]

---

## [Editor Report · Decision Letter 4]

17 Feb 2021

Dear Dr Woodford, 

On behalf of my colleagues and the Academic Editor, Lorenz Von Seidlein, I am pleased to inform you that we have agreed to publish your manuscript "Positron emission tomography and magnetic resonance imaging in experimental human malaria to identify organ-specific changes in morphology and glucose metabolism: a prospective cohort study" (PMEDICINE-D-20-04576R4) in PLOS Medicine.

PRESS

Sincerely, 

Beryne Odeny 

Associate Editor 

PLOS Medicine